# Exposure–Response Analysis of the Sodium–Glucose Cotransporter-2 Inhibitors Dapagliflozin and Empagliflozin on Kidney Hemodynamics in Patients with Type 2 Diabetes

**DOI:** 10.3390/jpm13050747

**Published:** 2023-04-27

**Authors:** Sjoukje van der Hoek, Jeroen V. Koomen, Erik J. M. van Bommel, Charlotte M. Mosterd, Rosalie A. Scholtes, Anne C. Hesp, Jasper Stevens, Daniel H. van Raalte, Hiddo J. L. Heerspink

**Affiliations:** 1Department of Clinical Pharmacy and Pharmacology, University of Groningen, University Medical Center Groningen, Hanzeplein 1, 9713 GZ Groningen, The Netherlands; sjoukje_vdhoek@hotmail.com (S.v.d.H.); j.stevens@umcg.nl (J.S.); 2Department of Anesthesiology, University of Groningen, University Medical Center Groningen, Hanzeplein 1, 9713 GZ Groningen, The Netherlands; j.v.koomen@umcg.nl; 3Department of Endocrinology and Metabolism, Amsterdam University Medical Centers, Location VUMC, Diabetes Center, De Boelelaan 1118, 1081 HZ Amsterdam, The Netherlands; e.v.bommel@jbz.nl (E.J.M.v.B.); c.m.mosterd@amsterdamumc.nl (C.M.M.); r.scholtes@amsterdamumc.nl (R.A.S.); a.c.hesp@amsterdamumc.nl (A.C.H.); d.vanraalte@amsterdamumc.nl (D.H.v.R.); 4Amsterdam Cardiovascular Sciences, VU University, De Boelelaan 1108, 1081 HZ Amsterdam, The Netherlands

**Keywords:** SGLT2 inhibitors, dapagliflozin, empagliflozin, pharmacokinetics, pharmacodynamics, renal hemodynamics, type 2 diabetes, response variability

## Abstract

Sodium–glucose cotransporter-2 (SGLT2) inhibitors improve markers for renal and cardiovascular outcomes in patients with and without type 2 diabetes (T2D). To assess whether individual differences in plasma drug exposure can explain inter-individual response variation, we characterized the exposure–response relationship for two SGLT2 inhibitors on several clinical and kidney hemodynamic variables. Data were obtained from two studies, RED and RECOLAR, assessing the effects of once-daily 10 mg dapagliflozin or empagliflozin, respectively, on kidney hemodynamics in patients with T2D. Individual plasma exposure was estimated using non-compartmental analyses and exposure–response relationships were assessed using linear mixed-effects models. In 23 patients participating in RED, the dapagliflozin geometric mean apparent area under the concentration-time curve during one dosing interval at steady state (AUC_0–tau,ss_) was 1153.1 µg/L*h (coefficient of variation (CV) 81.8%) and associated, per doubling, with decreases in body weight (0.29 kg, *p* < 0.001), systolic blood pressure (0.80 mmHg, *p* = 0.002), measured glomerular filtration rate (mGFR) (0.83 mL/min, *p* = 0.03), and filtration fraction (0.09%, *p* = 0.04). In 20 patients participating in RECOLOR, the empagliflozin geometric mean AUC_0–tau,ss_ was 2035.7 nmol/L*h (CV 48.4%) and associated, per doubling, with decreases in body weight (0.13 kg, *p* = 0.002), systolic blood pressure (0.65 mmHg, *p* = 0.045), and mGFR (0.78 mL/min, *p* = 0.002). To conclude, dapagliflozin and empagliflozin plasma exposure was highly variable between patients and associated with inter-individual variation in response variables.

## 1. Introduction

Sodium–glucose cotransporter-2 (SGLT2) inhibitors increase urinary glucose excretion and decrease body weight, blood pressure, and urinary albumin excretion [1,2]. Importantly, SGLT2 inhibitors also reduce the risk of cardiovascular events, heart failure, and kidney outcomes in patients with and without type 2 diabetes (T2D). These clinical benefits appear to be largely independent of improvement in glycemic control [3,4,5,6,7,8,9].

A generally accepted mechanism underlying the benefits on kidney outcomes of SGLT2 inhibitors is that these agents restore tubuloglomerular feedback and thereby reduce glomerular hyperfiltration through dilation of the efferent arteriole (in type 2 diabetes) or constriction of the afferent arteriole (in hyperfiltering type 1 diabetes) [1,10,11]. This reduction in glomerular hyperfiltration is clinically manifested by an acute reduction in estimated glomerular filtration rate (eGFR) and is associated with a slower eGFR decline during prolonged treatment compared to placebo, indicating kidney preservation [12]. Two mechanistic studies suggested that the SGLT2 inhibitors dapagliflozin and empagliflozin acutely reduce intraglomerular pressure and GFR in patients with T2D and preserved kidney function [13,14].

Despite these clinical benefits at a population level, there is a large inter-individual variation in response to SGLT2 inhibitors, and, consequently, a considerable proportion of patients remain at increased risk of cardiovascular disease and progressive kidney function loss. The albuminuria-lowering response, an established surrogate endpoint for kidney disease progression in clinical trials, was unsatisfactory in approximately 20% of patients [15,16]. Furthermore, the response on albuminuria is reproducible after re-exposure, suggesting a true pharmacological response rather than random variation [15]. The underlying mechanisms of this individual variation in response to SGLT2 inhibition are incompletely understood. Systemic exposure to several SGLT2 inhibitors is variable between patients and related to responses in metabolic variables including glycemic control, body weight, and systolic blood pressure [17,18,19,20]. Less is known about the association between exposure to SGLT2 inhibition and responses in kidney variables including kidney hemodynamic changes. We, therefore, performed a post hoc analysis of two SGLT2 clinical studies, Renoprotective Effects of Dapagliflozin in Type 2 Diabetes (RED) and Renohemodynamic Effects of Combined empagliflozin and LosARtan (RECOLOR) [13,14], to investigate the association between systemic exposure to the SGLT2 inhibitors dapagliflozin and empagliflozin and changes in kidney and kidney hemodynamic variables. 

## 2. Materials and Methods

### 2.1. Study Design

Dapagliflozin exposure and response data were obtained from 24 participants in the dapagliflozin arm of the RED study (ClinicalTrials.gov ID: NCT02682563) (Figure 1A). The study design and primary results have been published previously [13]. In short, the RED study was a double-blind, randomized study that compared the effects of treatment with dapagliflozin 10 mg once daily to the sulfonylurea derivative gliclazide 30 mg once daily for 12 weeks on kidney hemodynamics in 44 patients with T2D. Main inclusion criteria were age 35 to 75 years, glycated hemoglobin A1c (HbA1c) 48 mmol/mol to 75 mmol/mol (6.5% to 9%), metformin monotherapy, well-controlled blood pressure and, if indicated, a stable, maximum tolerable dose of a renin–angiotensin system (RAS) inhibitor. The most important exclusion criteria were an eGFR < 60 mL/min/1.73 m^2^, urine albumin-to-creatinine ratio (UACR) > 300 mg/g, a cardiovascular event within 6 months before inclusion, and use of nonsteroid anti-inflammatory drugs or diuretics that could not be discontinued 3 months before and during the intervention period. The primary endpoints were changes in measured GFR (mGFR) and effective renal plasma flow (ERPF) from baseline to week 12 of treatment with dapagliflozin versus gliclazide. Secondary endpoints included changes in body weight, systolic blood pressure, fasting plasma glucose (FPG), HbA1c, and urinary glucose excretion (UGE), among others.

Empagliflozin exposure and response data were obtained from the empagliflozin treatment period in the RECOLAR study (ClinicalTrials.gov ID: NCT04238702) (Figure 1B). The primary results have been published previously [14]. The RECOLAR study was a randomized, double-blind, placebo-controlled, 4-period cross-over mechanistic intervention study to assess the kidney hemodynamic effect of mono- and combination therapy with empagliflozin and the angiotensin blocker losartan in 24 patients with T2D. The main inclusion criteria were age 45 to 80 years, HbA1c 48 mmol/mol to 86 mmol/mol (6.5% to 10%), and treatment with metformin and/or a sulfonylurea derivative. The most important exclusion criteria were an eGFR < 60 mL/min/1.73 m^2^, UACR > 300 mg/g, hypertension that could not be controlled other than with alpha-blockers and/or beta-blockers, chronic use of an SGLT2- or RAS inhibitor, use of nonsteroid anti-inflammatory drugs or diuretics that could not be discontinued during the study and a cardiovascular event within 6 months before inclusion. Study participants were randomized to four 1-week double-dummy treatment periods with placebo, losartan 50 mg/daily, empagliflozin 10 mg/daily or losartan 50 mg/daily, and empagliflozin 10 mg/daily with 4 weeks wash-out periods in between. At the end of every treatment period, mGFR and ERPF were assessed as primary endpoints. Secondary endpoints were, among others, body weight, systolic blood pressure, FPG, and UGE. For the current analysis, all participants in the RECOLAR study were included and data from the period in which they received a placebo were used as baseline values, and data from the period in which they received empagliflozin were used to assess exposure and response (Figure 1B).

Both clinical studies were performed in accordance with the Declaration of Helsinki and Good Clinical Practice and the study protocols were approved by the local medical ethics committees. All participants signed written informed consent before any study-specific procedure commenced.

### 2.2. Pharmacokinetic Measurements

In the RED study, pharmacokinetic measurements were performed on the 10 timed blood samples that were primarily taken for the kidney hemodynamic measurements at the end of the treatment period at week 12. Participants were instructed to take the last dose of dapagliflozin at 8 p.m. the evening before the study visit, and dapagliflozin plasma concentrations were determined in samples collected, according to protocol, between 12 h and 21 h after dosing. Plasma samples were stored at −80 °C until shipment to the Department of Clinical Pharmacy and Pharmacology of the University Medical Center in Groningen. Dapagliflozin analysis was performed using a validated liquid chromatography with tandem mass spectroscopy method (calibration range of 1–500 μg/L, a lower limit of quantification (LLOQ) of 1 µg/L, intra-assay precision (coefficient of variation (CV)) of 1.1–13.7%, inter-assay precision of 0.0–1.6% and accuracy of 94.6–101.0%) [21]. In the RECOLAR study, pharmacokinetic measurements were performed on the four timed blood samples primarily taken for the kidney hemodynamic assessments at the end of the treatment period with empagliflozin and placebo. Participants were instructed to take the last dose of empagliflozin at 10 p.m. the evening before the study visit, and empagliflozin concentrations were determined in samples collected, according to protocol, between 10 h and 13 h after dosing. Plasma samples were stored at −80 °C until shipment to the Department of Clinical Pharmacy and Pharmacology of the University Medical Center in Groningen. Concentrations of empagliflozin were measured using a validated liquid chromatography with tandem mass spectroscopy method (calibration range of 2–1000 μg/L, LLOQ of 2 µg/L, intra-assay precision (CV) of 1.0–9.9%, inter-assay precision of 1.1–3.7% and accuracy of 94.6–101.6%) [21].

### 2.3. Pharmacodynamic Measurements

In both studies, mGFR and ERPF were, respectively, assessed by inulin or iohexol and para-aminohippuric acid clearances with timed blood and urine sampling as described previously [22,23]. The filtration fraction (FF) was obtained by dividing mGFR by ERPF. In the RED study, the kidney hemodynamic assessments were performed at baseline and at the end of the treatment period at week 12. For the current analysis, we used the measurements obtained during clamped euglycemia, minimizing the effects of hyperglycemia on kidney and systemic hemodynamics. During this procedure, plasma glucose levels were maintained at 5 mmol/L using an insulin infusion of 40 mU/min/m^2^ body surface area and adjusting the rate of 20% glucose infusion, with supplementary glucose infusion when plasma glucose levels decreased by >0.2 mmol/L in 5 min. In the RECOLAR study, the kidney hemodynamics were obtained at the end of all four treatment periods in every participant, in a fasted state. The measurements at the end of the period, in which participants took only a placebo, were defined as the reference, i.e., baseline, values, and the measurements at the end of the treatment period with empagliflozin and placebo as response values. In both studies, 24 h urine collections to quantitate glucose excretion were obtained the day before the visits for the kidney assessments. Fasting blood samples were drawn at the start of these visits and used to assess drug-induced change in glucose and HbA1c (RED study only). Systolic blood pressure was measured using an automated oscillometric device (Dinamap, GE Healthcare, Little Chalfont, UK) at the brachial artery of the nondominant arm at the start of both visits. Measurements were performed in triplicate at 1- to 2-min intervals and the mean of the last two measurements was used. All participants were asked to adhere to a “normal” salt (9–12 g or 150–200 mmol per day) and protein (1.5–2.0 g/kg per day) intake the week before the kidney hemodynamic measurements.

### 2.4. Estimation of Individual Plasma Exposure to Dapagliflozin and Empagliflozin

The individual, apparent, plasma exposure to dapagliflozin and empagliflozin, defined as the area under the plasma curve during one dosing interval (AUC_0–tau,ss_) at steady state, was estimated using non-compartmental analysis. The elimination rate constant and apparent volume of distribution were estimated from the individual natural log-transformed concentration-time profiles. Repeated linear regression analysis was performed on the last data points to determine the best fit of the curve through the terminal part of the curves, as assessed by the adjusted R^2^, by including at least three data points and provided the slope was negative. As pharmacokinetic samples were collected at a steady state, a correction for residual drug from the prior dose was conducted by subtracting concentration measurements by the estimated concentration at T = 24 h following first-order elimination kinetics. Based on the apparent volume of distribution and the elimination rate constant, apparent AUC_0–tau,ss_ was obtained by dividing the drug dose by the individual clearance.

### 2.5. Evaluation of Association between Demographics and Plasma Exposure

Associations between age, sex, body weight, eGFR, mGFR, UACR, serum albumin, gamma-glutamyl transferase (GGT), alanine aminotransferase (ALAT), and, if available, alkaline phosphatase (AF), and aspartate aminotransferase (ASAT), and plasma drug exposure were evaluated using linear regression analysis. Non-normally distributed variables, as assessed by visual inspection, were log-transformed with base 2 to better approximate a normal distribution. Variables with a *p*-value < 0.10 in univariate analysis were selected for the development of a multivariable model. The multivariable model was developed using forward inclusion (*p*-value of <0.10) and model comparison was based on an F-test.

### 2.6. Evaluation of the Association between Exposure and Response

Response variables of interest in both studies were body weight, systolic blood pressure, FPG, UGE, mGFR, ERPF and FF, and HbA1c additionally in the RED study. In the RED study, measurements performed at the baseline visit were used as baseline values and compared to the values at the end of the 12 weeks treatment period. In the RECOLAR study, the measurements at the end of the period in which participants used only placebo, were defined as baseline values and compared to the values at the end of the 1-week treatment period with empagliflozin (and placebo). Urinary glucose values were log-transformed with base 2 before analyses to approximate a normal distribution. Differences in the variables of interest between baseline and end of treatment visit were assessed using a linear mixed-effect model with a random intercept per individual and visit as a fixed effect. 

The relationship between drug exposure in terms of apparent AUC_0–tau,ss_ and response was assessed using linear mixed-effects models. In all exposure–response analyses, a log-transformation with base 2 of AUC_0–tau,ss_ was applied to better approximate a normal distribution. Exposure to the drug was assumed to be zero at baseline. As the base model, a model with a random intercept per individual was fitted to the data. This model was compared to a random intercept model including apparent AUC_0–au,ss_ as a fixed effect to assess whether exposure can explain inter-individual variation. Model comparison was performed using a likelihood ratio test and a *p*-value < 0.10 was considered a significant improvement in model performance. A *t*-test was performed to evaluate whether the fixed effect estimate of apparent AUC_0–tau,ss_ was significantly different from zero. All linear mixed-effects models were fitted using full maximum-likelihood estimation in R version 3.6.3 using package nlme (version 3.1-144).

## 3. Results

The baseline characteristics of the included participants are presented in Table 1. Participants in the RECOLAR study were slightly older and had a longer diabetes duration compared to the participants in the RED study. The populations were comparable regarding body weight, systolic blood pressure, HbA1c, and GFR.

### 3.1. Dapagliflozin Exposure

In total, 237 plasma samples were available for quantification of the dapagliflozin plasma concentration in 24 participants in the RED study. In one participant, all 10 dapagliflozin concentrations were below the LLOQ. This participant was excluded from the analysis due to poor study drug adherence. Two other samples were below LLOQ and excluded from the analysis, resulting in the inclusion of 225 dapagliflozin concentrations in the exposure analysis. The demographics of the 23 included participants are presented in Table 1. The geometric mean apparent AUC_0–tau,ss_ after 10 mg dapagliflozin was 1153.1 µg/L*h (CV 81.8%).

### 3.2. Empagliflozin Exposure

In total, 84 plasma samples were available for quantification of the empagliflozin plasma concentration in 21 of the 24 included subjects in the RECOLAR study. One subject was excluded from the pharmacokinetic analysis because no negative slope could be obtained in the non-compartmental analysis and therefore estimation of the apparent AUC_0–tau,ss_ was considered not reliable. The demographics of the 20 included participants are presented in Table 1. The geometric mean apparent AUC_0–tau,ss_ after 10 mg empagliflozin was 2035.7 nmol/L*h (CV 48.4%).

### 3.3. Evaluation of Association between Demographics and Plasma Exposure

For dapagliflozin, female sex (β = −57.72% change in AUC, *p* = 0.06), GGT (β = −0.44% change in AUC per % increase, *p* = 0.049), and ALAT (β = −2.43% change in AUC per unit/L increase, *p* = 0.06) were associated with exposure in the univariate analysis (Table 2). Modeling both sex and ALAT resulted in the most optimal multivariable model fit with females having an estimated 49.89% lower AUC compared to males and each unit/L higher ALAT being associated with a 1.94% lower AUC, albeit these associations did not reach statistical significance when they were both entered in the multivariable model (both *p* = 0.13) (Table 2). For empagliflozin, body weight (β = −1.68% change in AUC per kg increase, *p* = 0.007), eGFR (β = −2.60% change in AUC per mL/min/1.73 m^2^ increase, *p* = 0.003), UACR (β = 0.21% change in AUC per % increase, *p* = 0.04), and serum albumin (β = 11.64% change in AUC per g/L increase, *p* = 0.052) were associated with exposure in the univariate analysis (Table 2). In the multivariable model, body weight (β = −1.39% change in AUC per kg increase, *p* = 0.008), eGFR (β = −1.41% change in AUC per unit increase, *p* = 0.06), and serum albumin (β = 10.36% change in AUC per g/L increase, *p* = 0.02) were associated with empagliflozin exposure (Table 2).

### 3.4. Evaluation of the Association between Exposure and Treatment Response

A significant change in all variables was observed after dapagliflozin treatment in the RED study (Table 3). After the empagliflozin treatment, UGE, FPG, body weight, and mGFR changed significantly in the RECOLAR study (Table 3). For dapagliflozin, it was estimated that every doubling of apparent AUC_0–tau,ss_ was associated with an increase in UGE of 48.7% (*p* < 0.001), a decrease in body weight of 0.29 kg (*p* < 0.001), in systolic blood pressure of 0.80 mmHg (*p* = 0.002), in HbA1c of 0.51 mmol/mol (*p* < 0.001), in FPG of 0.09 mmol/L (*p* = 0.002), in mGFR of 0.83 mL/min (*p* = 0.03) and in FF of 0.09% (*p* = 0.04) (Table 3, Figure 2). No significant relation between exposure to dapagliflozin and ERPF was observed (β = −2.69 mL/min, *p* = 0.06). For empagliflozin, it was estimated that every doubling of apparent AUC_0–tau,ss_ was associated with an increase in UGE of 48.1% (*p* < 0.001), a decrease in body weight of 0.13 kg (*p* = 0.002), in systolic blood pressure of 0.65 mmHg (*p* = 0.045), in FPG of 0.16 mmol/L (*p* < 0.001) and in mGFR of 0.78 mL/min (*p* = 0.002) (Table 3, Figure 3). No significant relation between exposure to empagliflozin and ERPF (β = 0.69 mL/min, *p* = 0.85) or FF (β = −0.13%, *p* = 0.22) was observed (Table 3).

## 4. Discussion

In this post hoc analysis of two clinical studies with dapagliflozin and empagliflozin, we showed that both dapagliflozin and empagliflozin exposure is highly variable between patients with type 2 diabetes. The variation in systemic exposure to these SGLT2 inhibitors seems to be associated with clinical variables previously observed with SGLT2 inhibitor exposure including demographic, anthropomorphic, metabolic, kidney, and liver function variables.

The inter-individual variation in plasma exposure to dapagliflozin and empagliflozin was high with a coefficient of variation of, respectively, 82% and 48%. For dapagliflozin, part of this variation was explained by sex, with women having an approximately 50% lower apparent AUC_0–tau,ss_ compared to men. This is opposite to what has been previously reported [24]. We do not have a clear explanation for our findings, but we cannot rule out a chance finding due to the relatively small number of women in our study. High levels of liver enzymes (GGT and ALAT) also showed a trend towards a lower dapagliflozin exposure, which is surprising because impaired liver function would be expected to be associated with higher drug exposure due to decreased metabolism [25]. However, in our cohort liver function variables were within a relatively narrow and near-normal range, and none of the participants had known or apparent liver disease. Therefore, GGT and ALAT do not reflect altered liver function. Prior studies demonstrated a strong association between GFR and dapagliflozin exposure [24,26], but we did not find this in our study possibly due to the narrow and normal GFR range. For empagliflozin, part of the inter-individual variation in plasma exposure was explained by differences in body weight, eGFR, and plasma albumin. In keeping with the existing literature, lower body weight and eGFR were associated with higher exposure [20,27]. Furthermore, higher levels of albumin were associated with higher drug exposure, also consistent with the reported higher exposure associated with higher serum protein, of which albumin is the main constituent [20]. No effect of sex on exposure was detected, probably because there were only two females enrolled.

Exposure–response relationships for SGLT2 inhibitors have been well characterized for UGE and glycemic control, as primary efficacy variables, with higher exposure being consistently associated with a larger increase in UGE and decreases in FPG and HbA1c [17,20,28,29,30]. The observed strength of the association between dapagliflozin exposure and UGE and FPG changes was in the same order of magnitude as previously reported in pooled analyses of 13 dapagliflozin studies [17,19]. Besides lowering glucose, SGLT2 inhibitors impact several other clinical variables for which however the exposure–response relationships have been less extensively studied. The effect of dapagliflozin exposure on body weight and systolic blood pressure has been assessed in patients with T2D and in patients with CKD without diabetes [17,18,19]. In contrast to the CKD study [18], we did find a significant association between exposure and body weight change. The relation between SGLT2 inhibitors exposure and mGFR response has only been published, to the best of our knowledge, for dapagliflozin in a CKD population without diabetes [18]. Similar to patients with CKD and without diabetes we also observed clear associations between exposure and kidney hemodynamic variables. For empagliflozin, exposure–response relations have only been described for UGE and glycemic control [20,28,31]. The association between empagliflozin exposure and UGE we observed seems a bit stronger than reported before [28]. However, due to the use of different models and outcome variables this, and the association for FPG, is hard to compare with the other studies. In addition to glycemic variables, we now demonstrated that empagliflozin exposure is also associated with changes in systolic blood pressure, body weight, and mGFR.

This study shows that a lower exposure is associated with a poorer response and suggests that in non-responders to dapagliflozin or empagliflozin a higher exposure may be considered to improve the patient’s response. As only one dose level was tested in the current analysis, we cannot speculate on whether higher dosages lead to improved outcomes. For that, additional analysis should be performed, characterizing dose-exposure–response relationships for multiple dose levels. 

Despite exposure–response relationships between the RED and RECOLAR studies appearing to be similar, we do not recommend comparing these associations head-to-head due to important differences in the enrolled populations and study designs. In the RED study, 65% of the participants were using a RASi versus none in the RECOLAR study. Furthermore, the herein-reported kidney hemodynamics in the RED study were measured during euglycemic clamp while in the RECOLAR study in a fasted state, which complicates direct comparisons. Lastly, the treatment duration differed between the studies with a one-week treatment duration in RECOLAR and 12 weeks in RED. The hemodynamic response with SGLT2 inhibitors is fully present after one week of treatment as shown previously but one week of treatment is insufficient for a full response to body weight. This can also explain the smaller-than-expected body weight reduction with empagliflozin. 

Our study has several limitations, mostly related to the study populations. Both studies included relatively homogeneous populations, especially regarding body weight, liver, and kidney function. It is, therefore, possible that we could not identify the relationship between these variables and plasma drug exposure very accurately. In addition, no participants with CKD were included, which further limits the generalizability of the associations between GFR and exposure. Because few females were enrolled in both studies, the association between sex and dapagliflozin but not empagliflozin exposure should be interpreted in the context of low statistical power, and additional studies are required. Lastly, the actual times of drug intake and sample collection were not recorded; therefore, we must assume that this was undertaken as close to the protocol as possible.

## 5. Conclusions

The plasma exposure to dapagliflozin and empagliflozin showed a large variation between patients and related to inter-individual variation in mGFR response as well as in response to several clinical variables.

## Figures and Tables

**Figure 1 jpm-13-00747-f001:**
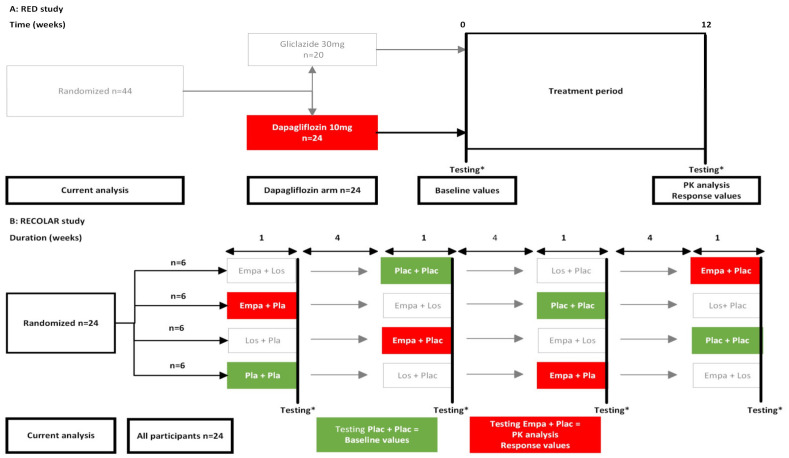
Study designs RED (**A**) and RECOLAR (**B**) study with data used for current analysis. (**A**) Participants were randomized to treatment with dapagliflozin or gliclazide. For the current analysis, participants randomized to dapagliflozin were included, data from testing at week 0 served as baseline values, and data from testing at week 12 were used to obtain exposure and response. (**B**) Cross-over design in which participants were randomly assigned to 1 of 4 sequences of the 4 treatment periods. For the current analysis, all participants were included and data from the placebo-placebo period served as baseline values, and data from the empagliflozin-placebo period were used to obtain exposure and response. Plac = placebo, Los = losartan 50 mg, Empa = empagliflozin 10 mg; * Testing: includes body weight, blood pressure, blood and urine assessments, and kidney hemodynamics.

**Figure 2 jpm-13-00747-f002:**
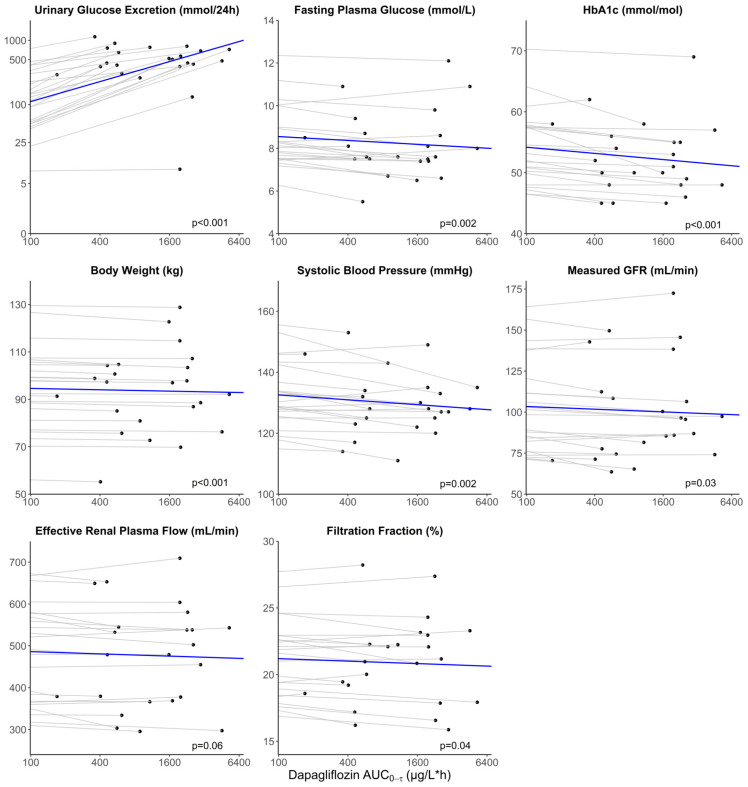
Relationship between apparent dapagliflozin plasma exposure and response after 12 weeks. Observations (black circles) of individual responses are plotted versus the apparent area under the plasma concentration-time curve (AUC_0–tau,ss_). The dotted gray lines represent the individual change versus AUC_0–tau,ss_. The solid blue line represents the population change vs AUC_0–tau,ss_. Note the start of the x-axis at AUC = 100 µg/L*h and the different starting points of the y-axis.

**Figure 3 jpm-13-00747-f003:**
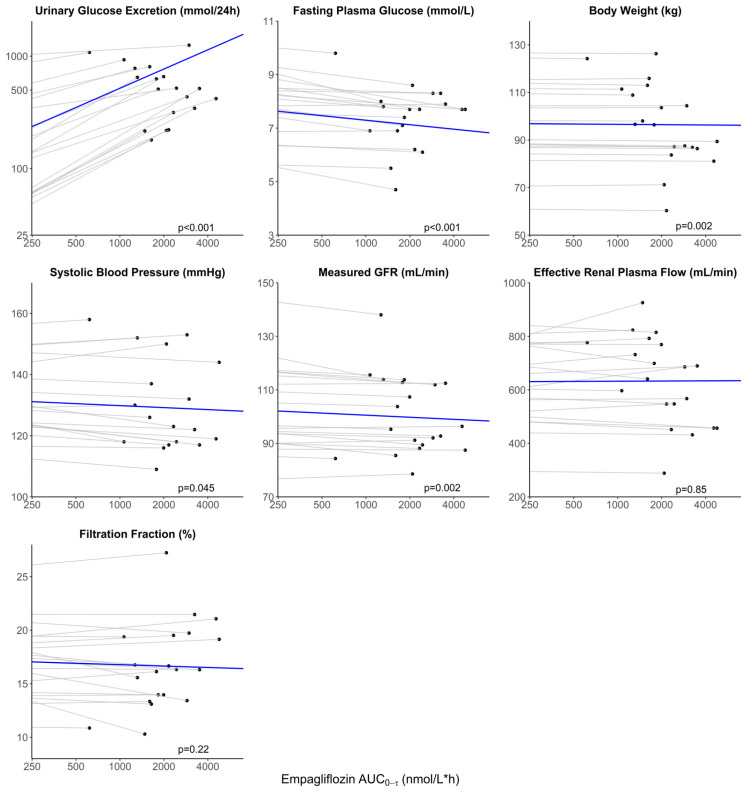
Relationship between apparent empagliflozin plasma exposure and response after 1 week. Observations (black circles) of individual responses are plotted versus the apparent area under the plasma concentration-time curve (AUC_0–tau,ss_). The dotted gray lines represent the individual change versus AUC_0–tau,ss_. The solid blue line represents the population change vs. AUC_0–tau,ss_. Note the start of the x-axis at AUC_0–tau,ss_ = 250 nmol/L*h and the different starting points of the y-axis.

**Table 1 jpm-13-00747-t001:** Baseline characteristics participants in RED and RECOLAR studies.

	RED (*n* = 23)	RECOLAR (*n* = 20)
Age (yr)	63 ± 7	66 ± 6.3
Female sex, n (%)	5 (22)	2 (10)
Body weight (kg)	96.6 ± 18.3	97.9 ± 16.4
Body mass index (kg/m^2^)	30.8 ± 4.0	31.1 ± 4.0
Systolic blood pressure (mmHg)	137.9 ± 13.9	136.2 ± 9.7 ^†^
Hemoglobin (mmol/L)	8.5 ± 0.7	8.5 ± 0.9
mGFR (mL/min)	113.6 ± 20.0	108.4 ± 21.1
eGFR (CKD-EPI) (mL/min/1.73 m^2^)	84.8 ± 13.7	87.4 ± 11.8
HbA1c (mmol/mol)	57.6 ± 7.2	57.3 ± 9.9
Albumin (g/L)	38.0 ± 2.5	37.5 ± 2.0
Alkaline phosphatase (U/L)	76.6 ± 21.6	NA
Gamma-glutamyl transferase (U/L)	33.0 [19.5, 53.0]	31 [22.0, 50.5]
Aspartate aminotransferase (U/L)	21.0 [17.3, 26.0]	21 [19.3, 28.3] ^†^
Alanine aminotransferase (U/L)	30.6 ± 14.5	NA
Albuminuria (mg/24 h)	11.0 [7.5, 16.9]	12.1 [6.1, 22.0]
UACR (mg/mmol)	0.81 [0.51, 1.16]	1.18 [0.66, 1.44]
Diabetes duration (yr)	9.7 ± 4.2	12.5 ± 5.7
Metformin, n (%)	23 (100)	20 (100)
Sulfonylurea derivative, n (%)	0	9 (45)
RAS inhibitor, n (%)	15 (65.2)	0 (0.0)

Abbreviations: CKD-EPI, Chronic Kidney Disease Epidemiology Collaboration; eGFR/mGFR, estimated/measured glomerular filtration rate; HbA1c, Glycated hemoglobin A1c; IQR, interquartile range; RAS, renin–angiotensin system; SD, standard deviation; UACR, urine albumin-to-creatinine ratio; Data are presented as mean (SD), median [IQR] or n (%); ^†^ Two participants with missing values.

**Table 2 jpm-13-00747-t002:** Patient characteristics associated with apparent AUC_0–tau,ss_ of dapagliflozin and empagliflozin.

	RED (*n* = 23)				RECOLAR (*n* = 20)			
	Dapagliflozin 10 mg				Empagliflozin 10 mg			
	Univariate		Multivariable		Univariate		Multivariable	
	β (% Change)	*p*	β	*p*	β (% Change)	*p*	β	*p*
Age (yr)	−1.37	0.63			2.47	0.18		
Female sex	−57.72	0.06	−49.89	0.13	−13.43	0.71		
Body weight (kg)	0.55	0.62			−1.68	0.01	−1.39	0.008
eGFR (CKD-EPI) (mL/min/1.73 m^2^)	−1.05	0.47			−2.60	0.003	−1.41	0.06
mGFR (mL/min)	−0.31	0.77			−0.73	0.18		
UACR (%)	−0.025	0.89			0.21	0.04		
HbA1c (mmol/mol)	−0.57 ^†^	0.73			0.94	0.44		
Serum albumin (g/L)	−6.41	0.40			11.64	0.052	10.36	0.002
Alkaline phosphatase (U/L)	−1.14	0.21			NA			
Gamma-glutamyl transferase (%)	−0.44 ^†^	0.049			0.015 ^†,‡^	0.93		
Aspartate aminotransferase (%)	−0.56 ^†^	0.36			0.35 ^†^	0.30		
Alanine aminotransferase (U/L)	−2.43	0.06	−1.94	0.13	NA			

Abbreviations: AUC_0–tau,ss,_ area under the concentration-time curve during one dosing interval at steady state; CKD-EPI, Chronic Kidney Disease Epidemiology Collaboration; HbA1c, glycated hemoglobin A1c; eGFR/mGFR, estimated/measured glomerular filtration rate (CKD-EPI); NA, not available; UACR, urine albumin-to-creatinine ratio; Note: AUCs were log-transformed before analyses, therefore β represent % change in AUC per unit increase in variable, except were stated per % increase in variable (^†^); ^‡^ Two participants with missing values; RED: the adj R^2^ of the multivariable model is 0.18 (*p* = 0.056); RECOLAR: the adj R^2^ of the multivariable model is 0.60 (*p* < 0.001).

**Table 3 jpm-13-00747-t003:** Changes in response variables and their relationship with plasma drug exposure.

	RED Study (*n* = 23)						RECOLAR Study (*n* = 20)					
	Dapagliflozin 10 mg						Empagliflozin 10 mg					
	Baseline	End of Treatment	Change	*p*	β per Doubling AUC	*p*	Baseline	End of Treatment	Change	*p*	β per Doubling AUC	*p*
Body weight (kg)	96.6 ± 18.3	93.6 ± 17.3	−3.0 (−4.0, −2.0)	<0.001	−0.29 (−0.39, −0.19)	<0.001	97.9 ± 17.1	96.6 ± 17.1	−1.3 (−2.0, −0.6)	0.002	−0.13 (−0.20, −0.06)	0.002
SBP (mmHg)	137.9 ± 13.9	129.8 ± 10.6	−8.1 (−12.9, −3.2)	0.003	−0.80 (−1.27, −0.34)	0.002	136.2 ± 9.7 ^†^	130.3 ± 14.9 ^‡^	−6.1 (−12.3, 0.04)	0.06	−0.65 (−1.26, −0.03)	0.045
HbA1c (mmol/mol)	57.6 ± 7.2	52.3 ± 5.9	−5.3 (−7.5, −3.0)	<0.001	−0.51 (−0.73, −0.29)	<0.001	NA	NA	NA	NA	NA	NA
FPG (mmol/L)	9.20 ± 1.54	8.20 ± 1.55	−1.00 (−1.5, −0.5)	<0.001	−0.09 (−0.14, −0.04)	0.002	9.4 ± 2.6	7.4 ± 1.2 ^‡^	−1.6 (−2.2, −1.0)	<0.001	−0.16 (−0.22, −0.01)	<0.001
UGE (mmol/24 h)	7.6 [0.8–39.7]	480 [391–703]	409 (170, 974) ^§^	<0.001	48.69 (35.88, 62.71)	<0.001	6.1 [0.8, 60.9]	517 [331, 719] ^‡^	477 (212, 1017) ^§^	<0.001	48.05 (36.82, 60.20)	<0.001
mGFR (mL/min) ^¶^	109.4 ± 27.8	100.1 ± 30.4	−9.3 (−16.5, −2.0)	0.01	−0.83 (−1.53, −0.12)	0.03	108.4 ± 21.1	100.5 ± 14.8	−7.9 (−12.2, −3.6)	0.002	−0.78 (−1.22, −0.33)	0.002
ERPF (mL/min) ^¶^	506.7 ± 124.7	474.3 ± 122.6	−32.3 (−60.4, −4.3)	0.03	−2.69 (−5.46, 0.08)	0.06	623.2 ± 169.6	634.7 ± 164.5	11.5 (−60.9, 83.9)	0.75	0.69 (−6.65, 8.03)	0.85
FF (%)	21.8 ± 2.9	20.9 ± 3.3	−1.0 (−1.9, −0.04)	0.04	−0.09 (−0.18, −0.005)	0.04	18.2 ± 4.1	16.7 ± 4.0	−1.5 (−3.5, 0.6)	0.16	−0.13 (−0.34, 0.08)	0.22

Abbreviations: AUC, area under the concentration-time curve; CI: confidence interval; ERPF, effective renal plasma flow; FF, filtration fraction; FPG, fasting plasma glucose; HbA1c, glycated hemoglobin (A1c); IQR, interquartile range; mGFR, measured glomerular filtration rate; NA: not available; SBP: systolic blood pressure; SD, standard deviation; UGE: urinary glucose excretion; Data are presented as mean ± SD, mean (95%CI) or median [IQR]; Change between baseline and end of treatment and their *p* values are based on linear mixed-effects modeling; Beta values are expressed as absolute change in variable per doubling of apparent AUC (95% CI), except for urinary glucose (% change); ^†^ Two participants with missing value; ^‡^ One participant with missing value; ^§^ Analysis performed after log-transformation; ^¶^ In RED study measured during euglycemia in RECOLAR study in a fasted state.

## Data Availability

The data that support the findings of this study are available from Daniel van Raalte upon reasonable request.

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
