# Peer review of "Exposure–Response Analysis of the Sodium–Glucose Cotransporter-2 Inhibitors Dapagliflozin and Empagliflozin on Kidney Hemodynamics in Patients with Type 2 Diabetes"

_jpm, 2023, doi:10.3390/jpm13050747_

Round 1

Reviewer 1 Report

Van der Hoek and colleagues have tried to answer to a very crucial question of daily clinical practice, i.e. finding a reason for the interindividual response to SGLT2-inhibitors (SGLT2-i) treatment. 

The association between SGLT2-i exposure and changes in kidney and kidney hemodynamic variables have been retrospectively assessed in two studies, RED and RECOLOR, using dapagliflozin and empagliflozin, respectively.

The manuscript is well written and the statistical analysis is well described and performed.

However, there are some major critical points:

- The design of the study should be different in order to find the answer to the initial question. It should be prospective, enrolling subjects at different stages of kidney function (with CKD too), with or without exposure to metformin.

- There is no mention to the reason why canagliflozin, the first SGLT2-i with a demonstrated beneficial effect on kidney outcomes, has not be included in the analysis.

- The sample size is limited, not allowing a generalizability of the findings.

- It is not clear wheter the subjects enrolled in RECOLAR and teated with losartan were excluded from the analysis.

- Does the variation of plasma exposure to dapagliflozin and empagliflozin able to determined a limitation of the effect reduction on A1c and pre-treatment estimated reduction of eGFR in a long-term observation?

Reviewer 2 Report

Summary Statement

van der Hoek et al. examined whether inter-individual response variation to SLGT2i therapy is due to differences in plasma drug exposure. The authors performed a post-hoc analysis using data collected from two clinical trials: RED (Dapagliflozin) and RECOLAR (Empagliflozin).  They employed non-compartmental, linear regression, univariate, and multivariate analyses to address this overall question.

Strengths

The study is appropriate for the journal.

The research question is clearly identified and clinically relevant.

The manuscript is clear and well-written. The authors do a nice job of delineating, consolidating, and comparing and contrasting pertinent information (i.e., inclusion and exclusion criteria, primary and secondary endpoints, study protocol, etc.) from the RED vs. RECOLAR studies.

The methods appear sound.

Data are clearly presented.

The authors do not overstate their findings and fully acknowledge and describe several key limitations to the study.

Areas for Improvement

Major Points

The sample size in each of the analysis groups is quite low, although the authors concede to this issue in their discussion.

Minor Points

Ten of the 25 references are works that include the corresponding author (Heerspink).

Author Response

Response to Reviewer 2 comments

Comments and Suggestions for Authors

 Summary Statement

van der Hoek et al. examined whether inter-individual response variation to SLGT2i therapy is due to differences in plasma drug exposure. The authors performed a post-hoc analysis using data collected from two clinical trials: RED (Dapagliflozin) and RECOLAR (Empagliflozin). They employed non-compartmental, linear regression, univariate, and multivariate analyses to address this overall question.

Strengths

The study is appropriate for the journal.

The research question is clearly identified and clinically relevant.

The manuscript is clear and well-written. The authors do a nice job of delineating, consolidating, and comparing and contrasting pertinent information (i.e., inclusion and exclusion criteria, primary and secondary endpoints, study protocol, etc.) from the RED vs. RECOLAR studies.

The methods appear sound.

Data are clearly presented.

The authors do not overstate their findings and fully acknowledge and describe several key limitations to the study.

Response:

We thank the reviewer for valuing our article and recognizing the clinical importance of our exposure-response analysis of dapagliflozin and empagliflozin on kidney hemodynamics. Please find our point-to-point response below. Changes in our manuscript with respect to the previous version are indicated with track changes.

Comments

Major Points

Comment 1: The sample size in each of the analysis groups is quite low, although the authors concede to this issue in their discussion.

Response 1: We agree with the reviewer the limitation of the relatively small sample sizes of both studies. In the discussion, we therefore acknowledge the limited generalizability of our findings, also due to the relatively homogeneous populations and relatively few females were enrolled. However, the RED and RECOLAR studies are the only studies available assessing the effects of SGLT2 inhibition on renal hemodynamics and we believe therefore that our findings regarding large differences between patients in plasma exposure to dapagliflozin and empagliflozin and its relation to especially mGFR response contribute to a better understanding of inter-individual response variability to SGLT2 inhibitors and are thus relevant to be reported.  

Minor Points

Comment 2: Ten of the 25 references are works that include the corresponding author (Heerspink).

Response 2: We agree with the reviewer that our citations were unbalanced and therefore have added the following references to the manuscript:

  1. Fioretto, P.; Zambon A.; Rossato M.; Busetto L., Vettor R. SGLT2 Inhibitors and the Diabetic Kidney. Diabetes Care 2016, 39 Suppl 2, 165. 10.2337/dcS15-3006.
  2. The EMPA-KIDNEY Collaborative Group; Herrington W.G.; Staplin N.; Wanner C.; Green J.B.; Hauske S.J.; Emberson J.R.; Preiss D.; Judge P.; Mayne K.J. et al. Empagliflozin in Patients with Chronic Kidney Disease. N Engl J Med 2023, 388, 117-127. 10.1056/NEJMoa2204233.
  3. Solomon, S.D.; McMurray J.J.V.; Claggett B.; de Boer R.A.; DeMets D.; Hernandez A.F.; Inzucchi S.E.; Kosiborod M.N.; Lam C.S.P.; Martinez F. et al. Dapagliflozin in Heart Failure with Mildly Reduced or Preserved Ejection Fraction. N Engl J Med 2022, 387, 1089-1098. 10.1056/NEJMoa2206286.
  4. Kim, N.H., Kim N.H. Renoprotective Mechanism of Sodium-Glucose Cotransporter 2 Inhibitors: Focusing on Renal Hemodynamics. Diabetes Metab J 2022, 46, 543-551. 10.4093/dmj.2022.020
  5. Kasichayanula, S.; Liu X.; Pe Benito M.; Yao M.; Pfister M.; LaCreta F.P.; Humphreys W.G., Boulton D.W. The influence of kidney function on dapagliflozin exposure, metabolism and pharmacodynamics in healthy subjects and in patients with type 2 diabetes mellitus. Br J Clin Pharmacol 2013, 76, 432-444. 10.1111/bcp.12056.
  6. Macha, S.; Mattheus M.; Halabi A.; Pinnetti S.; Woerle H.J., Broedl U.C. Pharmacokinetics, pharmacodynamics and safety of empagliflozin, a sodium glucose cotransporter 2 (SGLT2) inhibitor, in subjects with renal impairment. Diabetes Obes Metab 2014, 16, 215-222. 10.1111/dom.12182.

Reviewer 3 Report

The current manuscript submitted by Sjoukje van der Hoek et al and titled “Exposure-response analysis of the sodium-glucose cotrans-2 porter-2 inhibitors dapagliflozin and empagliflozin on kidney 3 hemodynamics in patients with type 2 diabetes”. Authors evaluated the differences in circulating drug exposure and their response relationship on clinical and hemodynamic variable in human individuals with type-2 diabetes. These individuals treated with 2-different SGLT2 inhibitors Dapa and Empa to accesses their response relationship and effects on clinical parameters.  Though it is mentioned that the current clinical study has several limitation includes lower number of study population together with limited gender variation, mainly time of drug intake and samples collection. The study design is appropriate and well conducted, written all the methodology details and results described very well. However, several minor concerns authors may have considered. 

Minor comments:

1) In both study populations (RED & RECOLAR) administrated with Dapa and Empa, Is there any relationship of 24 h urinary glucose excretion with fasting plasma glucose?

2) Is authors observed any dose response for urine glucose excretion?

3) In Table2, urinary albumin ratios (uACR) or Albuminuria can be added.

4) It would be very interesting if authors determined any inflammatory markers in plasma and kidney injury markers in urine like KIM1, NGAL in these study population and relationship in response to drug exposure?

5) An optional or not a mandatory, that It would easy to fallow and appreciated if authors could add representative or schematic diagram for study design and methodology/drug expose response.

Round 2

Reviewer 1 Report

The authors replied appropriately to the comments.

However, regarding "kidney hemodynamic assessments in the RED trial were performed during euglycemic clamp and therefore minimizing the effects of hyperglycemia on kidney hemodynamics", the authors should be take into account that also hyperinsulinemia, even in a short term, is able to modify kidney hemodynamics. It should be interesting to have some comments on this point in the revised manuscript.